# Burn-Induced Cardiac Mitochondrial Dysfunction via Interruption of the PDE5A-cGMP-PKG Pathway

**DOI:** 10.3390/ijms21072350

**Published:** 2020-03-28

**Authors:** Jake J. Wen, Claire B. Cummins, Ravi S. Radhakrishnan

**Affiliations:** Division of Pediatric Surgery, Department of Surgery, University of Texas Medical Branch, Galveston, TX 77555, USA; cbcummin@utmb.edu

**Keywords:** burn injury, mitochondria, oxygen consumption, electron transport chain, sildenafil

## Abstract

Burn-induced heart dysfunction is a key factor for patient mortality. However, the molecular mechanisms are not yet fully elucidated. This study sought to understand whether burn-induced heart dysfunction is associated with cardiac mitochondrial dysfunction and interruption of the PDE5A-cGMP-PKG pathway. Sixty percent total body surface area (TBSA) scald burned rats (±sildenafil) were used in this study. A transmission electron microscope (TEM), real-time qPCR, O2K-respirometer, and electron transport chain assays were used to characterized molecular function. Cardiac mitochondrial morphological shapes were disfigured with a decline in mitochondrial number, area, and size, resulting in deficiency of cardiac mitochondrial replication. Burn induced a decrease in all mitDNA encoded genes. State 3 oxygen consumption was significantly decreased. Mitochondrial complex I substrate-energized or complex II substrate-energized and both of respiratory control ratio (RCRs) were decreased after burn. All mitochondrial complex activity except complex II were decreased in the burn group, correlating with decreases in mitochondrial ATP and MnSOD activity. Sildenafil, a inhibitor of the PDE5A-cGMP-PKG pathway, preserved the mitochondrial structure, respiratory chain efficiency and energy status in cardiac tissue. Furthermore, sildenafil treatment significantly restored ADP-conjugated respiration in burned groups. In conclusion, cardiac mitochondrial damage contributes to burn-induced heart dysfunction via the PDE5A-cGMP-PKG pathway.

## 1. Introduction

Burns result in greater than 500,000 people seeking medical attention, over 40,000 hospitalizations, and around 4000 deaths in the United States each year [1]. The annual expense for treating burn patients is greater than $18 billion [2,3]. Burn-induced cardiodynamic derangements contribute to multiple organ system failure, sepsis, and death [4,5]. Higher total body surface area (TBSA) results in higher mortality in both human patients [6] and animals [7] due to cardiac dysfunction post burn [8,9,10].

In cardiomyocytes, there is growing evidence that stimulation of the cyclic GMP (cGMP)–protein kinase-G (PKG) axis dulls the cardiac stress responses, leading to attenuation of pathological hypertrophy, protection against ischemic injury, and enhanced cell survival [11]. One means of potentiating the cGMP–PKG axis is by inhibiting the degradation of cytosolic cGMP by phosphodiesterase-5 (PDE5). cGMP is the second messenger of several important signaling pathways based on distinct guanylate cyclases (GCs) in the cardiovascular system [12]. PKG1α arbitrates the down-stream effects that are involved in keeping the contractile force of cardiomyocytes and phosphorylates serine and threonine residues on numerous cytosolic proteins [13]. Seya et al. found that the NO-cGMP-PKG pathway might play a role in ischemic pre-conditioning and antioxidant cardiac protection via cardiac mitochondria [14], even though the precise mechanism remains to be discovered. PDE5 inhibitors, such as sildenafil (SIL), prolong the effect of NO, a potent vasodilator, and cGMP, causing increased vasodilation and smooth muscle tone regulation [15]. While the effects of sildenafil were previously studied in an in vivo burn model [16], its role on cardiac function has not been reported. The aim of this study was to reveal the role of the NO-cGMP-PKG1α pathway in burn-induced cardiac mitochondrial dysfunction.

Mitochondria are universal organelles that play key roles in bioenergetics and oxygen consumption to maintain homeostasis and normal organ function. As a result of severe burn, increased mitochondrial uncoupling contributes to the hypermetabolic stress response [17]. Additionally, burn-induced mitochondrial damage in heart tissues is correlated with increased oxidative stress [18,19,20]. While severe burns alter mitochondrial function in tissues, such as skeletal muscle [21,22,23] and adipose tissue [24,25,26], mitochondrial function in heart tissue after burn has not been well studied. In our previous study, we observed that burn induced cardiac dysfunction and cardiac mitochondrial damage occur via the AMPK-SIRT1-PGC1α-NFE2L2-ARE signaling pathway [27]. However, the role of PDE5A-cGMP-PKG pathway as well as the relationship between PDE5A-cGMP-PKG pathway and cardiac mitochondrial responses remain unclear. In this study, we sought to reveal whether the PDE5A-cGMP-PKG pathway and mitochondrial damage contribute to burn-induced heart dysfunction, and, whether the administration of sildenafil would be helpful in maintaining burn-induced cardiac mitochondrial damage.

## 2. Results

### 2.1. Cardiac Mitochondrial Structure and Morphology

To explore the effect of burn injury on cardiac mitochondria ultrastructure, we utilized transmission electron microscopy (TEM). TEM images showed a disruption of mitochondrial distribution and polymorphism of the mitochondrial ultrastructure after burn (Figure 1A.a,A.c). The morphometric analysis revealed a 36% decreased number of mitochondria in burn group (vs. sham, Figure 1B.a). The area and size of mitochondria dramatically reduced 40% and 44%, respectively, in the burn group (vs. sham; Figure 1B.b,B.c). No effect of sildenafil on sham control was observed (Figure 1A.b), but sildenafil partially recovered burn-induced cardiac mitochondrial disruption (Figure 1A.d), mitochondrial numbers (Figure 1B.a), mitochondrial area (Figure 1B.b), and mitochondrial size (Figure 1B.c). These data suggest that burn injury results in the derangement of cardiac mitochondrial morphology and sildenafil treatment is effective in preserving myocardial mitochondrial ultrastructure after burn.

### 2.2. Cardiac mitDNA Replication

To analyze mitDNA replication after burn, cardiomyocyte gDNA was extracted and qPCR was employed to measure mitDNA copy number by normalization of nuclear GAPDH/β-actin. D-loop gene (Figure 2A) and mitDNA encoded genes, including ND1 (complex I; Figure 2B), COX II (complex IV; Figure 2C), and ATP6 (complex V; Figure 2D), copy numbers were significantly decreased by 83%, 86%, 70%, and 87%, respectively, after burn but were normalized by sildenafil administration (Figure 2A–D). These data suggest that the derangement of cardiac mitochondrial morphology resulted in declines of mitDNA replication and that sildenafil treatment is effective in preserving myocardial mitDNA replication.

### 2.3. mtDNA-Encoded Gene Expression in Burned Group

To determine if burn-induced derangement of mitochondrial superstructures was correlated with mitDNA-encoded gene expressions, we used a quantitative assessment of the expressions of mtDNA encoded genes. The data demonstrated a decrease (62–83%) in the expression of mtDNA encoded genes, which are necessary for the assembly of functional complex 1 (*ND1, ND2, ND3, ND4, ND5, and ND6*; Figure 3A), complex III (*CYTB*; Figure 3B), complex IV (*COXI*, *COX II*, and *COXIII*; Figure 3C), and complex V (*ATP6* and *ATP8*; Figure 3D) after burn. Sildenafil treatment normalized all gene expressions except *ND4* (Figure 3Aa–Ac,Ae,Af,B–D). The expression of *ND4* after treatment with sildenafil was improved by 45% in comparison to the burn group, but it did not reach statistical significance (Figure 3Ad). Sildenafil didn’t interfere with mitDNA-encoded gene expressions (Figure 3). This suggests that burn-induced cardiac mitochondrial abnormal structure and morphology is secondary to decreases in mitochondrial replication and mitDNA encoded genes.

### 2.4. Cardiac Mitochondrial Function

To determine if alteration of mitochondrial structure and declines of mitDNA gene expression affect mitochondrial function, we evaluated the rate of oxygen consumption. There was no significant change in state 2 respiration (Figure A1A). The state 3 respiration driven with complex I substrates (P+G+M) and complex II substrates (S+R), was reduced by 73–76% after burn (Figure 4A.a,A.b). To study mitochondrial integrity, addition of cytochrome c after OXPHOS resulted in a 2.5-fold increase of respiration in the burn group (Figure 4A.c), indicating that severe burn-induced mitochondrial electron transport chain damage was associated with interruption of mitochondrial integrity. Uncoupled reagent carbonyl cyanide m-chlorophenyl hydrazone (CCCP) was used to test mitochondrial maximum respiration; burn injury resulted in a 64% lower maximal mitochondrial ATP production (Figure 4A.d). Sildenafil treatment completely restored the maximal mitochondrial ATP production to control level (Figure 4A). Respiratory control ratios (RCR) were decreased in both complex I and complex III respiration after burn by 53% and 48%, respectively (Figure 4B.a,B.b). Sildenafil treatment attenuated the loss in complex I and complex III energized respiration as well as the RCRs in cardiac mitochondria after burn (Figure 4A.a,A.b,B.a,B.b). These data suggest that severe burned-related compromise of mitochondrial function was, completely or at least partially, ameliorated by sildenafil treatment.

### 2.5. Cardiac Mitochondrial Electron Transport Chain Activity

To determine whether burn-induced mitochondrial dysfunction would affect mitochondrial electron transport chain activities, mitochondrial complex activity assays were employed. Electron transport chain efficiencies were decreased by 45% for complex I, 67% for complex III, 89% for complex IV, and 57% for complex V (Figure 5A.a–A.d), respectively. No significant change was seen in complex II activity (Figure A1B). This led to metabolic energy deficiency and decreased mitochondrial antioxidant (MnSOD) activity in myocardial tissue. These data indicate that myocardial ATP content after burn was decreased by 52% and MnSOD activity was decreased by 22% (Figure 5B,C). Treatment with sildenafil increased respiratory complex activity (Figure 5A), MnSOD activity (Figure 5C), and tissue ATP levels (Figure 5B).

## 3. Discussion

Previous results have demonstrated that the NO-PDE5-cGMP-PKG pathway preserves heart function and cardiomyocyte mitochondrial function through PKG1α kinase activation [28] and that PDE5A is upregulated in the hypertrophied heart [29,30]. We therefore hypothesized that the PDE5-cGMP-PKG pathway plays a very important role in burn-induced cardiac dysfunction via mitochondrial damage. We further hypothesized that sildenafil, a PDE5 inhibitor, would be effective in reducing cardiac dysfunction after severe burns. Our data indicate that burn induced cardiac mitochondrial dysfunction, evidenced by disturbances in mitochondrial morphology, decreased mitochondrial ATP generation, decreased mitochondrial oxygen consumption, and decreased mitochondrial electron transport chain activities. Sildenafil treatment resulted in complete or partial resolution of burn-induced mitochondrial dysfunction. This work is the first study showing the involvement of the PDE5A-cGMP-PKG pathway in burn-induced mitochondrial dysfunction, and the effectiveness of sildenafil treatment.

Cardiac damage is a well-documented event that increases mortality and morbidity after severe burns [9]. Research in in vivo and in vitro models has demonstrated that poor cardiac contractility starts immediately after thermal injury, continues for approximately 36 h after burn, and resolves by 72 h after burn [9,31]. In this study, we found burn-induced cardiac mitochondrial damage evidenced by disruption of mitochondrial morphology and mitochondrial superstructures, altered activity of the electron transport chain, and decreased mitochondrial oxygen consumption resulting in ATP deficiency and decreased mitochondrial antioxidant enzyme activity (MnSOD).

Multiple other disease states are associated with cardiac mitochondrial superstructure changes, such as alcohol-induced heart failure [32,33], cardiac hypertrophy [34], cardiovascular reperfusion injury [35], congenital heart diseases [36], heart failure [37], myocardial infarction [38], and myocardial ischemia-reperfusion injury [39]. In this study, we showed that burn-induced heart dysfunction is associated with cardiac mitochondrial morphological damage. Our results show that burn induced changes in cardiac mitochondrial morphology, as evidenced by decreased of mitochondrial number, mitochondrial area, and mitochondrial size. This indicates that one mechanism of burn-induced heart dysfunction is through cardiac mitochondrial superstructure changes similar to other stress-induced cardiac mitochondrial structure changes seen in other disease states.

Cardiac mitochondrial morphological change has been correlated with mitochondrial electron transport chain dysfunction in other disease states such as cardiac hypertrophy [40], ischemia-reperfusion injury [41,42], and heart failure [40]. Our observations demonstrate that burn-induced mitochondrial state 3 oxygen consumptions were significantly decreased, leading to deficiencies of mitochondrial complex I and complex III activities. We saw no changes in state 2 respiration or complex II activity, which is consistent with previously published results on mitochondrial dysfunction after burns [19].

Declines in cardiac mitochondrial number and size correlate with mitochondrial replication dysfunction in heart failure [43], doxorubicin-Induced cardiomyopathy [44], and cardiac ischemia reperfusion injury [45]. Our observations demonstrate that burn injury resulted in a decline of cardiac mitDNA replication which resulted in burn-induced decline of cardiac mitochondrial number. Burn injury caused heart dysfunction by derangement of cardiac mitochondrial structure, inhibition of cardiac mitDNA replication, and interruption of cardiac mitochondrial electron transport chain function via inhibition of mitDNA-encoded gene expressions. Many disease-causing mutations affect mitochondrial replication factors, and a detailed understanding of the replication process may aid to explain the pathogenic mechanisms underlying a number of mitochondrial diseases [46]. Deficiency in mtDNA replication or nucleotide metabolism leads to point mutations, and/or depletion of mtDNA [47]. mitochondrial D-loop has surmised to play the roles in nucleoid organization, nucleotide homeostasis, and mtDNA replication [48]. Our observations indicated that mitochondrial structure was damaged, mitDNA was depleted, and mitochondrial D-loop copy number was significantly decreased after burn.

Current investigations indicate that the molecular mechanisms underlying burn-induced cardiac dysfunction are associated with persistent β-Adrenergic Receptors (β-ARs) stimulation, leading to increase NO levels [9]. However, little is known of the role of the NOS-NO-cGMP-PKG pathway in burn-induced myocardial dysfunction. Our results demonstrate that the administration of sildenafil, a PDE5 inhibitor, after burn resulted in improvement in all parameters of cardiac mitochondrial function. This includes mitochondrial morphology, mitDNA replication, mitDNA-encoded gene expressions, mitochondrial electron transport chain function, and mitochondrial complex activity. Burn injury increases catecholamine levels both in human patients [49,50,51] and in animals [52,53]. However, PDE5 inhibition improves catecholamine responsiveness in heart failure [54]. The PDE5A-cGMP-PKG pathway is potentially one of the mechanisms by which burn injury induces cardiac mitochondrial dysfunction.

Previously published findings from our group demonstrated that burn-induced heart dysfunction occurred via the AMPK-SIRT1-PGC1α-NFE2L2-ARE pathway. The use of an AMPK inhibitor resulted in worsening of cardiac mitochondrial function after burn. In contrast, use of an AMPK activator or a PGC1α activator has a powerful cardio-protective effect against cardiac mitochondrial dysfunction and left ventricular dysfunction after burn [27]. We surmise that interaction between the AMPK-SIRT1-PGC1α-NFE2L2-ARE pathway and the PDE5A-cGMP-PKG pathway may play a very important role in burn-induced heart dysfunction.

In summary, our novel observations provide the first evidence that burn-induced heart dysfunction occurs via cardiac mitochondrial dysfunction and PDE5A-cGMP-PKG pathway interruption. The PDE5A inhibitor sildenafil has a potent cardio-protective effect.

## 4. Material and Methods

### 4.1. Ethics Statement

All animal procedures followed the experimental animal use protocol under the NIH instructions. Our protocols were certified by the IACUC (Institutional Animal Care and Use Committee) at the University of Texas Medical Branch (UTMB), TX (Protocol number: 1509059, 31 August 2015)

### 4.2. Rats

Male Sprague-Dawley rats (wild-type) were bought from Harlan Laboratories (Indianapolis, IN, USA). Six to nine animals were used in each group. The rats were kept in animal house to adapt for one week before the procedure, maintained on a light-dark cycle (12 h:12 h; ~25 °C), and freely administered food and water during study. A well-established model for the induction of a 60% TBSA full-thickness burn was utilized [55,56]. Briefly, rats (300–350 g) were given analgesia (buprenorphine, 0.05 mg/kg) by subcutaneous injection and anesthetized with general anesthesia (isoflurane, 3–5%), then submerged in 95–100 °C water using a protective mold. The dorsum was exposed for 10 s and the abdomen was exposed for 2 s, reliably creating a 60% TBSA full-thickness burn. Subsequently, the burned group were resuscitated with lactated Ringer (LR) solution (40 mL/kg, i.p. ± sildenafil, 1.5 mg/kg body weight) and oxygen immediately. Analgesia (buprenorphine, 0.05 mg/kg) was as needed every 6 h after burn. Due to damage to nerve ending in full-thickness burns, additional analgesia is rarely required. At pre-determined time points, rats were humanely euthanized by bilateral thoracotomy under anesthesia (≥5% isoflurane), and tissue was collected to be stored at −80 °C. For the isolation of mitochondria, fresh heart tissues were minced, immersed in mitochondrial isolation buffer (Mitochondria Isolation Kit, cat# ab110168, Abcam, Cambridge, MA, USA) and processed in 4 h after harvest.

### 4.3. Transmission Electron Microscopy (TEM)

Rat heart tissue was harvested at 24 hpb and put in Ito’s fixative (0.05 M cacodylate buffer containing glutaraldehyde: 2.5%, CaCl_2_: 0.03%, formaldehyde: 1.25%, and trinitrophenol: 0.03% in, pH 7.3) for 1 h at RT and subsequently overnight at 4 °C. After washing 3× (10 min/time using 0.1 M cacodylate buffer), the sample was immersed in 1% osmium tetroxide (OsO_4_) for 1 h and stained with 2% uranyl acetate in 0.1 M maleate buffer, and then embedded in Poly/Bed 812 (Polysciences, Warrington, PA, USA) after dehydration in different concentration of ethanol (50% 1× for 10 min, 75% 1× for 10 min, 95% 1× for 10 min, and 100% 3× for 10 min/time). Sorvall MT-6000 ultramicrotome (RMC, Tucson, AZ, USA) was used to make ultrathin sections. The ultrathin sections were measured in a Philips 201 transmission electron microscope (TEM, Philips Electron Optics, Eindhoven, The Netherlands) at 60 kV.

### 4.4. Morphometric Analysis

A morphometric analysis was performed under blinded conditions by systematic uniform random sampling with the Fiji-Image J Software (NIH, Bethesda, MD, USA) using 20 randomly selected images. In TEM images, the number, area(s), and volume density of organelles were determined using the morphometric technique with a dot grid [57]. Mitochondrial types, mitochondria clusters, and intermitochondrial junctions (IMJ) were determined using the point counting method [58]. The coefficient of energy efficiency of mitochondria (CEEM) was defined as the product of the number of mitochondrial cristae and the area of mitochondria [59].

### 4.5. Mitochondrial Copy Number

The Qiagen DNeasy Blood & Tissue Kit (Qiagen, Germantown, MD, USA. Cat# 69504) was applied to extract genomic DNA following the manufacturer’s instructions. To estimate mtDNA copy number, the rat control-region (D-loop), ND1 of one of mitochondrial complex I (CI) gene, COX II of mitochondrial complex VI (CIV) gene, ATP 6 of mitochondrial complex V (CV) gene and nuclear GAPDH were amplified by qPCR using specific primers (Table 1). The ratio of mitochondrial copy number was presented as copy number of the mitochondrial genome to the copy number of the nuclear housekeeping gene (GAPDH).

### 4.6. Gene Expression Analysis

Ten mg of heart tissue was utilized to isolate/purified total RNAs using an RNeasy Mini Kit (Qiagen, Gaithersburg, MD, Cat# 74104) according to manufactural introduction. The purified RNAs were treated by DNase I, RNase-free (NEB Inc., Westlake, LA, USA, Cat# M0303S) to digest contaminated genomic DNA. For judging the integrity and overall quality of isolated RNAs, 2 µg RNAs were run on 1% native agarose gels. A spectrophotometer (The DU^®®®^ 700 UV/Visible, Beckman Coulter, Pasadena, CA, USA) was applied to measure RNA quantitative by determination of absorbance at 260 and 280 nm (OD**_260/280_** ratio ≥ 2, 1 OD_260_ Unit = 40 µg/mL RNA). Two µg RNA was utilized to make cDNA using a SuperScript^®®®^ III Reverse Transcriptase (Invitrogen, Waltham, MA, USA. Cat# 18080044) following company manual. The cDNA was used as a template to run qPCR using iCycler Thermal Cycler (Bio-Rad, Hercules, CA, USA) with SYBR-Green Supermix (SYBR Green Real-Time PCR Master Mixes, ThermoFisher, Sugar Land, TX, USA, Cat# 4309155). The primers used in the study are shown in Table 2. The threshold cycle (Ct) values of target genes were normalized to housekeeping GAPDH gene, and calculated as fold change [60].

### 4.7. Mitochondria Isolation

Cardiac mitochondria were isolated/purified using the MitoCheck^®®®^ Mitochondrial (Tissue,) Isolation Kit (Cayman, Ann Arbor, MI, USA. Cat# 701010) following the manufacturer’s instruction. Briefly, the heart tissue was washed, suspended in mitochondrial homogenization buffer to be homogenized, and spun at 680× *g* for 15 min; the supernatant was then transferred to a new tube and centrifuged again at 8800× *g* for 15 min. Mitochondrial pellets were washed twice-using mitochondrial homogenization buffer containing protease inhibitor (Abcam Cat# ab201111). mitochondrial protein concentration was determined by bicinchoninic acid (BCA) assay. Aliquoted mitochondrial pellets were stored −80 °C for further usage.

### 4.8. The Mitochondrial Oxidative Phosphorylation (OXPHOS) Activities

The mitochondrial OXPHOS activities were measured using MitoCheck Complex Activity Assay Kits (Cayman Chemical, Ann Arbor, MI, USA. Cat# 700930 for complex I, Cat# 700940 for complex II, Cat# 700950 for complex II/III, Cat# 700990 for complex IV, and Cat# 701000 for complex V) according to manufacturer’s protocols. Briefly, Complex I activity was determined by measuring the decreased rate of NADH oxidation at 340 nm; Complex II activity was determined as a decrease in absorbance at 600 nm over time; Complex III activity was determined by measuring the reduction of cytochrome c at 550 nm; Complex IV activity was determined by the oxidation rate of reduced cytochrome c at 550 nm; and Complex V activity was determined by the rate of NADH oxidation c at 340 nm.

### 4.9. Cardiac Mitochondria Function Analysis

All the procedures for measurement of freshly cardiac mitochondrial function in vivo were as stated by ours and others’ publications [61,62,63]. Briefly, ~10 mg heart pieces were immersed in pre-cold BIOPS buffer. Permeabilization of the cardiomyocyte membrane was done via transferring cardiac myofiber to 2 mL of MIR05 buffer containing saponin (50 μg/mL) for 30 min at 4 °C. About 2 mg of permeabilized cardiac myofiber were used for measuring mitochondrial respiration function with a pre-calibrated air saturated MIR05 buffer using Oroboros oxygraph-2k (O2K) system (Oroboros, Innsbruck, Austria) by the consecutive addition of substrates and uncouplers. The leak respiration (state 2) was achieved by the addition of 1.5 mM octanoyl-l-carnitine, 5 mM pyruvate, 10 mM glutamate, and 2 mM malate (P+G+M) into the O2K chamber; state 3 respiration driven by mitochondrial CI substrates was achieved by the titration of 5 mM ADP; and state 3 respiration driven by mitochondrial complex II (CII) substrate was achieved by the addition of 10 mM succinate. To test the competence of the mitochondrial outer membrane, 10 μM cytochrome c was added. Finally, uncoupled OXPHOS was achieved by the addition of 5 μM of carbonyl cyanide m-chlorophenylhydrazone (CCCP). All data were analyzed by Oroboros DatLab software.

### 4.10. Statistical Analysis

The experiments were carried out in triplicate. All data were analyzed using GraphPad Prism8 software (GraphPAd, San Diego, CA, USA) and presented as mean ± standard deviation. Data were measured if it was normally distributed. Normally distributed data were used to be statistically analyzed by one-way ANOVA with Tukey’s test. No normally distributed data were analyzed by Kruskal–Wallis (K–W) programs. Significance is presented by * vs. sham rats or ^&^24 hpb vs. 24 hpb/SIL-treated) (*^,&^
*p* < 0.05, **^,&&^
*p* < 0.01, ***^,&&&^
*p* < 0.001).

## Figures and Tables

**Figure 1 ijms-21-02350-f001:**
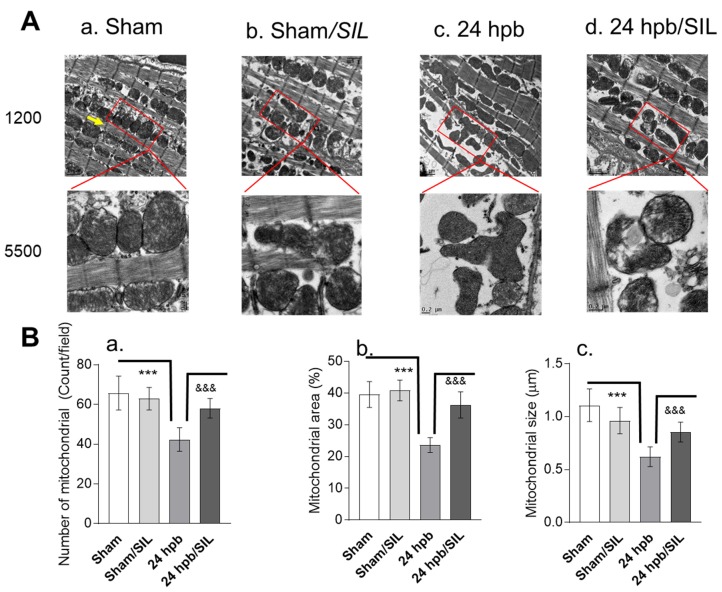
Analysis of left ventricle (LV) cardiomyocyte mitochondria of sham control rats and 24 h post burn (with/without sildenafil (SIL)). Sprague Dawley rats underwent 60% total body surface area (TBSA) scald burn with/without SIL, placed in regular animal room for 24 h, and were euthanized. (**A**), The representative transmission electron microscope (TEM) images from sham control group, sham/SIL group, 24 hpb group and 24 hpb/SIL group, showing even distributions of the mitochondria in the LV cardiomyocyte. The chain-like structure of the mitochondria is tightly connected with triad in the Z-line region (arrow). At 24 hpb group, the disruption of mitochondrial clusters and significant polymorphism of the mitochondrial ultrastructure were observed. Magnification 1200× and 5500×. (**B.a**), Means ± SEM of % of the mitochondrial area. (**B.b**), Means ± SEM of mitochondrial size. (**B.c**), Means ± SEM of mitochondrial clusters in cardiomyocyte and the number of mitochondria per cluster. Significance is shown as * (24 hpb vs. sham control) or **^&^** (24 hpb vs. 24 hpb/SIL), and presented as ***, **^&&&^**
*p* < 0.001 (*n* = ≥ 6 per group).

**Figure 2 ijms-21-02350-f002:**
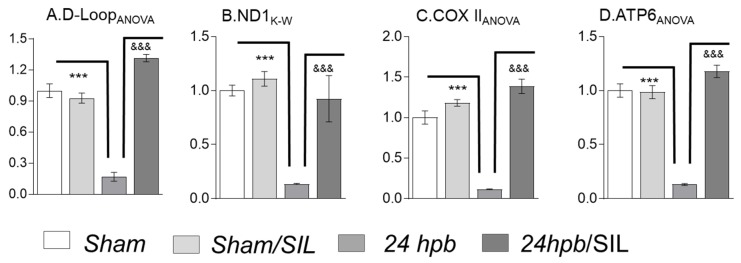
Analysis of cardiac mitochondrial replication and effects of SIL in burned Rats. Sprague Dawley rats underwent 60% TBSA scald burn and treated with sildenafil immediately after burn. heart tissues were collected at 24 hpb (24 hpb ± SIL). gDNAs were extracted using Dneasy Blood & Tissue Kits (Qiagen) and gene copy number was measured by quantitative PCR. Shown are the myocardial levels of mt D-Loop copy number (**A**), mtND1 copy number (**B**), mtCOX II copy number (**C**), and mtATP6 copy number (**D**). In all the figures, data are plotted as mean value ± SEM (*n* ≥ 8 rats per group). Significance is shown as * (24 hpb vs. sham control) or ^&^ (24 hpb vs. 24 hpb/SIL), and presented as ***, ^&&&^
*p* < 0.001 (*n* = ≥ 6 per group).

**Figure 3 ijms-21-02350-f003:**
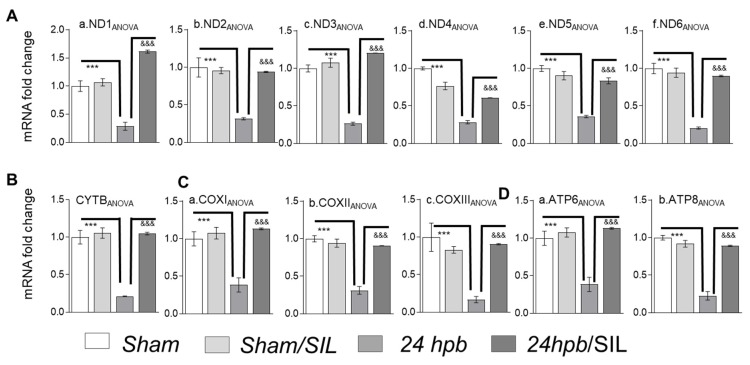
Analysis of mitDNA-encoded gene expression and Sildenafil effects in burned rats. Sprague Dawley rats underwent 60% TBSA scald burn and were treated with sildenafil. (**A**) Shown (panels a–f) are representatives of gene expressions for the mitDNA encoded complex I genes including. (**B**) Shown is mitDNA encoded complex III gene, *CYTB*. (**C**) Shown (panels a–c) are mitDNA encoded complex IV genes including *COXI, COXII and COXIII*. (**D**) Shown (panels a,b) are gene expressions of mitDNA encoded complex V genes including *ATP6* and *ATP8*. In all figures, data are presented as mean value ± SD. Significance is shown as * (24 hpb vs. sham control) or & (24 hpb vs. 24 hpb/SIL), and presented as ***, ^&&&^
*p* < 0.001 (*n* = ≥ 6 per group).

**Figure 4 ijms-21-02350-f004:**
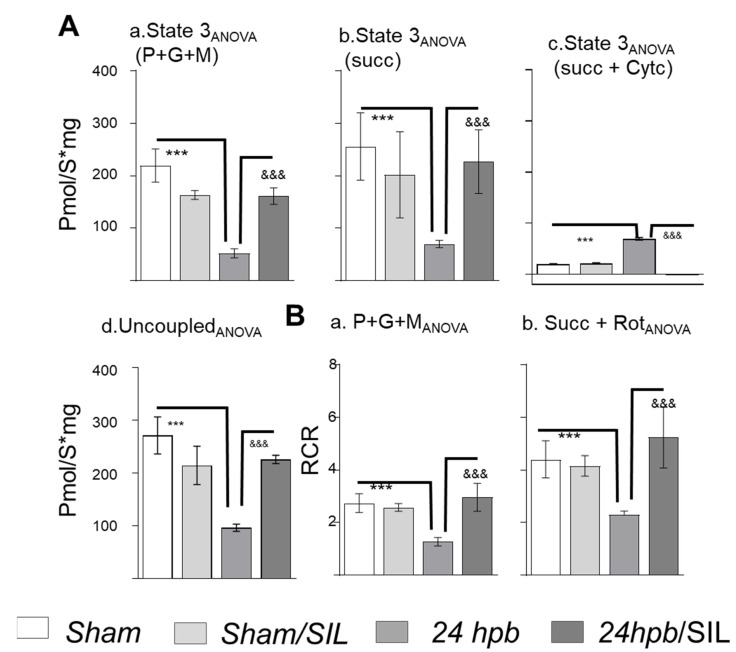
Analysis of mitochondrial respiration and Sildenafil effects in burned rats. Sprague Dawley rats underwent 60% TBSA scald burn and were treated with sildenafil. A small piece of tissue was collected at 24 h post burn (24 hpb ± SIL). (**A**) Electron flow and oxygen consumption were measured by using O2K respiratory system supported with orderly giving complex I substrates (glu/pyr/mal, panels a), complex II substrate (succinate + rotenone, panel b), cytchrome C (c) and uncoupled agent (CCCP) (d). (**B**) Respiratory control ratios energized by complex I substrate (RCR, panel a) and complex II substrate (panel b) were calculated as the ratios between state 3 and state 4 rates. In all figures, data are plotted as mean value ± SD. Significance is shown as * (24 hpb vs. sham control) or & (24 hpb vs. 24 hpb/SIL), and presented as ***, ^&&&^
*p* < 0.001 (*n* = ≥ 6 per group).

**Figure 5 ijms-21-02350-f005:**
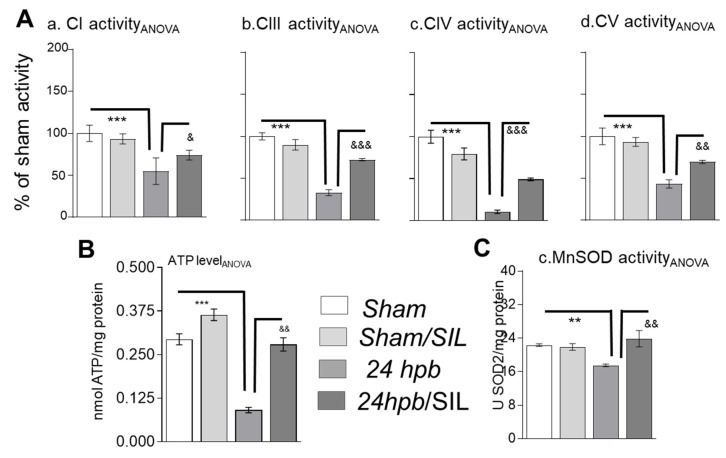
Analysis of mitochondrial complexes activities and Sildenafil effects in burned rats. Sprague Dawley rats underwent 60% TBSA scald burn and weretreated with sildenafil. A small piece of tissue was collected at 24 h post burn (24 hpb ± SIL). (**A**) heart tissues were collected at 24 h post burn (24 dpb ± SIL) and heart mitochondria were isolated by different speed centrifuges. Mitochondrial complex I (a), complex III (b), complex IV (c) and complex V (d) were measured by spectrophotometer. (**B**) Shown was representative of MIT ATP level in burned group. (**C**) Shown was representative of MIT MnSOD enzyme activity in burned group. In all the figures, the data are plotted as mean value ± SD. Significance is shown as * (24 hpb vs. sham control) or & (24 hpb vs. 24 hpb/SIL), and presented as ^&^
*p* < 0.05, **, ^&&^
*p* < 0.01, ***, ^&&&^
*p* < 0.001 (*n* = ≥ 6 per group).

**Table 1 ijms-21-02350-t001:** Used oligonucleotides for study of the mitochondrial copy number.

Gene Name	Forward Primer	Reverse Primer	Amplicon Size (bp)	Accession #
*B-actin*	ACTGGCATTGTGATGGACTC	GCTCGGTCAGGATCTTCATG	142	V01217.1
*D-Loop*	CGGATGCCTTCCTCAACATA	AGTCTTTCGAGCTTTGTCTATGA	107	KF011917.1
*ATP6*	GCACTAGCAGTACGACTAACAG	GTTGGTGGGCTGATGTCTATAA	101	KF011917.1
*COXII*	TCTCCCAGCTGTCATTCTTATTC	GCTTCAGTATCATTGGTGTCCTA	121	KF011917.1
*GAPDH*	ACTCCCATTCTTCCACCTTTG	CCCTGTTGCTGTAGCCATATT	105	NM_017008.4
*ND1*	CGCCTGACCAATAGCCATAA	CGACGTTAAAGCCTGAGACTAA	110	KF011917.1

**Table 2 ijms-21-02350-t002:** Used oligonucleotides for study of gene expression.

Gene Name	Forward Primer	Reverse Primer	Amplicon Size (bp)	Accession #
*B-actin*	ACAGGATGCAGAAGGAGATTAC	ACAGTGAGGCCAGGATAGA	117	V01217.1
*ATP6*	TAGGCTTCCGACACAAACTAAA	CTGCTAGTGCTATCGGTTGAATA	129	KF011917.1
*ATP8*	ATGCCACAACTAGACACAT	TTTGGGTGAGGGAGGTG	120	KF011917.1
*COXI*	GCCAGTATTAGCAGCAGGTATC	GGTGGCCGAAGAATCAGAATAG	125	KF011917.1
*COXII*	TCTCCCAGCTGTCATTCTTATTC	GCTTCAGTATCATTGGTGTCCTA	121	KF011917.1
*COXIII*	GCTGACCTCCAACAGGAATTA	CCTTCTATTAGGCTGTGATGGG	118	KF011917.1
*Cyt B*	CCTTCCTACCATTCCTGCATAC	TGGCCTCCGATTCATGTTAAG	118	KF011917.1
*GAPDH*	ACTCCCATTCTTCCACCTTTG	CCCTGTTGCTGTAGCCATATT	105	NM_017008.4
*ND1*	GGCTCCTTCTCCCTACAAATAC	AAGGGAGCTCGATTTGTTTCT	122	KF011917.1
*ND2*	CCCAACTATCACCACCATTCTC	TCGTGTTTGGGTCTGGTTAAG	79	KF011917.1
*ND3*	TTCTGCACGCCTTCCTTT	GGTTGTTTGAATCGCTCATGG	112	KF011917.1
*ND4*	GATGAGGCAACCAAACAGAAC	GTGTTGTGAGGGAGAGGATTAG	147	KF011917.1
*ND5*	GCCGCCACTATTATCTCCTTC	CTACTTCCTCCCACTCCATTTG	112	NM_133584.1
*ND6*	GGTGGGTTTGGATTGATTGTTAG	CCTCAGTAGCCATAGCAGTTG	148	NM_133584.1

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
