# Peer review of "Burn-Induced Cardiac Mitochondrial Dysfunction via Interruption of the PDE5A-cGMP-PKG Pathway"

_ijms, 2020, doi:10.3390/ijms21072350_

Round 1
Reviewer 1 Report
This draft described a well-designed, sophisticated experiment that demonstrated the effects of burning on cardiac mitochondria physiology and expression, and established the effectiveness of the sildenafil treatment in restoring the ADP-conjugated respiration. The data from replicated treatments are convincing. The writing is elegantly clear and pleasant to read. I'd recommend the acceptance of the draft for publication,pending any revision recommendation from other reviewers and editors.Author Response
Thanks very much and very appreciation for the reviewer’s inspiringly positive comments. We plan to continue to study the mechanisms behind burn-induced cardiac dysfunction. Future research may include clinical studies.
Reviewer 2 Report
The manuscript by Wen JJ et al. provides evidence that the PDE5A-cGMP-PKG pathway is involved in the burn-induced cardiac mitochondrial dysfunction, in a rat model. This conclusion is supported by the finding that sildenafil, a potent phosphodiesterase-5 inhibitor, reduces mitochondrial impairment in the treated animal group. Overall, the study widens the knowledge on the cardiac consequences of severe burns and identifies a possible therapeutic intervention aimed at preserving mitochondrial function in the myocardium. The work is of interest but the authors need to address several key aspects.
General comments
- it would be useful to include in the references the recent work by the Author’s group and to add some comments relating their previous recent study, in both the introduction and discussion.
- The aim of the study in the introduction, as well as the conclusion should better reflect the content of the manuscript
- The discussion can be shortened
- please check some typing errors
Specific comments.
- Introduction page 4 line 74-75: Authors should include some references at the end of this sentence. Although limited in number, there are very recent studies in the literature, as well as less recent ones that relate mitochondrial dysfunction in cardiomyocytes and severe burn injury.
- 2. Introduction page 4 line 78-80: The sentence must better reflect the aim of the study. Actually, the Authors do not evaluate the effect of the burn on cardiac function and specifically on cardiac hemodynamics (no data are provided on these parameters). Actually, the Authors sought to reveal the contribution of the PDE5A-cGMP-PKG pathway activation in triggering mitochondrial dysfunction in the myocardium after burn, and the effectiveness of the treatment with Sildenafil, as a PDE5A inhibitor, in reverting mitochondrial damage.
- Results page 4 subheading 2.1 and legend of figure 1: Mean values+/-SD are already shown in the bar graphs of Figure 1, to avoid repetition of data it could be better to include in the text the percentage variation of each parameter (see also comment n.7 on Figure 1 and the relative legend)
- Discussion page 7 lines: 163-165: please rephrase this sentence highlighting that this work is the first study showing (i) the involvement of the PDE5A-cGMP-PKG pathway in burn-induced mitochondrial dysfunction, and (ii) the effectiveness of Sildenafil treatment in inducing mitochondrial functional recovery. In fact, previous studies have already shown the morpho-functional damage of cardiac mitochondria after severe burn. In addition, other potential intracellular pathways have been recently described by the same Authors (J Am Coll Surg. 2020). Similar considerations apply to the conclusion, at the end of the discussion (page 9 lines 229-231).
- Methods page 10 paragraph 4.1. In this paragraph, the Authors should indicate the total number of animals used for the study, even if this information is reported in the figure legend. I have two more questions: (i) why waiting 6 hours after waking up the animals for the analgesic treatment? (ii) the method used for euthanasia should be described, "humanely euthanized" is not enough.
- Methods subheading 4.10, Statistical analysis: page 13 lines 340-341. The acronyms used in the text of this subheading to define the experimental groups are different from those reported in the figures and figure legends.
- Legend of Figure 1 page 29 last line. The indications related to the bar graphs B.a, B.b and B.c do not correspond to the parameters shown in the figure for each panel. Check also the correspondence between the symbols used in the figure to indicate significant differences and those reported in the legend.
- Legend of Figure 2 page 30 line line 706: …. (n>= 8 mice per group” probably rats??)
- Title of the legend of Figure 4 page 31: please insert “and” after respiration: “Analysis of mitochondrial respiration and Sildenafil effects……”
Author Response
Comments and Suggestions for Authors
The manuscript by Wen JJ et al. provides evidence that the PDE5A-cGMP-PKG pathway is involved in the burn-induced cardiac mitochondrial dysfunction, in a rat model. This conclusion is supported by the finding that sildenafil, a potent phosphodiesterase-5 inhibitor, reduces mitochondrial impairment in the treated animal group. Overall, the study widens the knowledge on the cardiac consequences of severe burns and identifies a possible therapeutic intervention aimed at preserving mitochondrial function in the myocardium. The work is of interest, but the authors need to address several key aspects.
A: Thanks very much for the reviewer’s overview evaluation that our work widens the knowledge in the field of burn-induced heart damage.
General comments
- it would be useful to include in the references the recent work by the Author’s group and to add some comments relating their previous recent study, in both the introduction and discussion.
A: Thank you for this suggestion! We added our presently published work in the Journal of the American College of Surgeons (JACS)[1]. Please look at our description in lines 76 to 80 of the introduction and lines 230-237 of the discussion.
- The aim of the study in the introduction, as well as the conclusion should better reflect the content of the manuscript
A: Thank you for this insight. We made changes to the introduction and the discussion to better reflect the content of the manuscript.
- The discussion can be shortened
A: This is a good suggestion and very much appreciated. We shortened our discussion part.
- please check some typing errors
A: Thank you for noticing this oversight. All typing errors have been corrected.
Specific comments.
- Introduction page 4 line 74-75: Authors should include some references at the end of this sentence. Although limited in number, there are very recent studies in the literature, as well as less recent ones that relate mitochondrial dysfunction in cardiomyocytes and severe burn injury.
A: Thank you for this suggestion. It’s a good point. We’ve added some more related important references in lines 74-75. Considering limited number of references regarding mitochondrial dysfunction in cardiomyocytes after burn, we tried to focus on only the relevant references regarding burn-induced cardiac mitochondrial dysfunction and cardiac mitochondrial dysfunction in other disease states.
- Introduction page 4 line 78-80: The sentence must better reflect the aim of the study. Actually, the Authors do not evaluate the effect of the burn on cardiac function and specifically on cardiac hemodynamics (no data are provided on these parameters). Actually, the Authors sought to reveal the contribution of the PDE5A-cGMP-PKG pathway activation in triggering mitochondrial dysfunction in the myocardium after burn, and the effectiveness of the treatment with Sildenafil, as a PDE5A inhibitor, in reverting mitochondrial damage.
A: We completely agree with the point and thank you for noting that we don’t mention or provide hemodynamics associated data! We changed our aim by replacing cardiac hemodynamics with cardiac mitochondrial damage.
- Results page 4 subheading 2.1 and legend of figure 1: Mean values+/-SD are already shown in the bar graphs of Figure 1, to avoid repetition of data it could be better to include in the text the percentage variation of each parameter (see also comment n.7 on Figure 1 and the relative legend)
A: Thank you for bringing this to our attention! We have revised our manuscript to avoid repetition in the specified areas.
- Discussion page 7 lines: 163-165: please rephrase this sentence highlighting that this work is the first study showing (i) the involvement of the PDE5A-cGMP-PKG pathway in burn-induced mitochondrial dysfunction, and (ii) the effectiveness of Sildenafil treatment in inducing mitochondrial functional recovery. In fact, previous studies have already shown the morpho-functional damage of cardiac mitochondria after severe burn. In addition, other potential intracellular pathways have been recently described by the same Authors (J Am Coll Surg. 2020). Similar considerations apply to the conclusion, at the end of the discussion (page 9 lines 229-231).
A: We appreciate this suggestion. We corrected the sentences following these suggestions. Additionally, we removed the sentence of page 9 lines 229-231.
- Methods page 10 paragraph 4.1. In this paragraph, the Authors should indicate the total number of animals used for the study, even if this information is reported in the figure legend. I have two more questions: (i) why waiting 6 hours after waking up the animals for the analgesic treatment? (ii) the method used for euthanasia should be described, "humanely euthanized" is not enough.
A: Thank you for this comment. We added the used animal numbers in the methods. To clarify our methods, analgesic treatment before the procedure (buprenorphine, 0.05 mg/kg) by subcutaneous injection and 3 -5% of isoflurane was used to anesthetize rats during the procedure. Additional analgesic was given as needed as indicated in the previously established protocol[2] to treat rats at every 6 hours after burn. Due to damage to nerve endings in full-thickness burns, additional analgesia is rarely required. This has been clarified in the methods section. Regarding the method of euthanasia, we added additional detail in the methods (Line 260).
- Methods subheading 4.10, Statistical analysis: page 13 lines 340-341. The acronyms used in the text of this subheading to define the experimental groups are different from those reported in the figures and figure legends.
A: Thank you for catching this. We revised the acronyms in the statistical analysis section.
- Legend of Figure 1 page 29 last line. The indications related to the bar graphs B.a, B.b and B.c do not correspond to the parameters shown in the figure for each panel. Check also the correspondence between the symbols used in the figure to indicate significant differences and those reported in the legend.
A: Thank you for this comment. We changed the legend to match the figures.
- Legend of Figure 2-page 30 line line 706: …. (n>= 8 mice per group” probably rats??)
Title of the legend of Figure 4 page 31: please insert “and” after respiration: “Analysis of mitochondrial respiration and Sildenafil effects……”
A: We apologize for this error. We corrected the mistake and added “and” after respiration.
- Wen JJ, Cummins CB, Szczesny B, Radhakrishnan RS (2020) Cardiac Dysfunction after Burn Injury: Role of the AMPK-SIRT1-PGC1alpha-NFE2L2-ARE Pathway. J Am Coll Surg. doi:10.1016/j.jamcollsurg.2019.12.029
- Johnson RE, Fudala PJ, Payne R (2005) Buprenorphine: considerations for pain management. J Pain Symptom Manage 29 (3):297-326. doi:10.1016/j.jpainsymman.2004.07.005
Round 2
Reviewer 2 Report
The Authors addressed all the issues.
Only one more thing, please look at the legend of Figure 1 and the symbols used in the bar graphs, they do not correspond.
Unless I have the old set of figures.
Author Response
Q: Only one more thing, please look at the legend of Figure 1 and the symbols used in the bar graphs, they do not correspond. Unless I have the old set of figures.
A: Thank you for noticing this oversight. We corrected it.
